# Frequency Limits of Sequential Readout for Sensing AC Magnetic Fields Using Nitrogen-Vacancy Centers in Diamond

**DOI:** 10.3390/s23177566

**Published:** 2023-08-31

**Authors:** Santosh Ghimire, Seong-joo Lee, Sangwon Oh, Jeong Hyun Shim

**Affiliations:** 1Quantum Magnetic Imaging Team, Korea Research Institute of Standards and Science, Daejeon 34113, Republic of Korea; santoshgmre@kriss.re.kr (S.G.); sj.lee@kriss.re.kr (S.-j.L.); sangwon.oh@kriss.re.kr (S.O.); 2Department of Applied Measurement Science, University of Science and Technology, Daejeon 34113, Republic of Korea

**Keywords:** AC sensing, sequential readout, AC frequency limit, nitrogen-vacancy color center, diamond

## Abstract

The nitrogen-vacancy (NV) centers in diamond have the ability to sense alternating-current (AC) magnetic fields with high spatial resolution. However, the frequency range of AC sensing protocols based on dynamical decoupling (DD) sequences has not been thoroughly explored experimentally. In this work, we aimed to determine the sensitivity of the ac magnetic field as a function of frequency using the sequential readout method. The upper limit at high frequency is clearly determined by Rabi frequency, in line with the expected effect of finite DD-pulse width. In contrast, the lower frequency limit is primarily governed by the duration of optical repolarization rather than the decoherence time (T${}_{2}$) of NV spins. This becomes particularly crucial when the repetition (dwell) time of the sequential readout is fixed to maintain the acquisition bandwidth. The equation we provide successfully describes the tendency in the frequency dependence. In addition, at the near-optimal frequency of 1 MHz, we reached a maximum sensitivity of 229 pT/$\sqrt{\mathrm{Hz}}$ by employing the XY4-(4) DD sequence.

## 1. Introduction

The sensing of alternating-current (AC) magnetic fields in the range of kilohertz to megahertz has been carried out in various applications, including nuclear magnetic resonance (NMR) [1], magnetic induction tomography [2], and magnetic communications [3]. Solid-state quantum spins, such as negatively charged nitrogen-vacancy (NV) centers in diamond [4,5,6], are among the most promising sensors for ac magnetic fields due to their high spatial resolution, which ranges from micro to nanometer [4,5,6,7,8,9,10,11,12,13,14,15]. The controllable density and depth of the quantum spins near the surface of host materials provide a reduced standoff distance from sources, resulting in such high resolution. In addition, the protocols for sensing ac magnetic fields allow control over both sensible frequency and bandwidth [16]. Moreover, the bandwidth that NV spins provide is often significantly wider than that of radio-frequency optically pumped magnetometers [17]. The boardband sensing using solid-state spin sensors enables the high-fidelity reception of rapidly varying or modulated ac signals.

AC-field sensing protocols called dynamical decoupling (DD) exploit a multiple number of phase-refocusing π pulses [16,18,19]. The stroboscopic pulse train periodically flips the quantum states and hence filters out a specific frequency, reducing the frequency window and detuning the quantum spins from surrounding magnetic noises. This increases the coherence time T${}_{2}$ of quantum spins [20,21,22] and thus enhances sensitivity to AC magnetic fields [5,23,24]. However, when detecting an oscillating signal with a time-varying or modulated envelope, such as nuclear spin precessions in NMR, the signal often persists longer than the T${}_{2}$ duration. To address this issue, the sequential readout (SR) scheme or quantum heterodyne (Qdyne) has been developed, based on the repetition of a block consisting of a DD sequence and a projective readout. The SR method works because the phase of the AC magnetic field continues even though the quantum states collapse during the projective readout [25,26,27]. In principle, the spectral resolution achievable with SR is limited only by the stability of external clocks synchronizing instrumentations, regardless of T${}_{2}$ [26]. This feature enabled the sensing of an AC magnetic field with sub-millihertz resolution [27] and the acquisition of high-resolution NMR spectra from micron-scale liquid samples [1,28,29].

The range of frequency to which the SR method is sensitive is certainly crucial for its applications in the field. As the sensing frequency increases, the time interval τ in Figure 1a decreases, and eventually the interrogation window closes. Considering the finite width of the π pulses, one can expect the Rabi-oscillation frequency to be the high-frequency limit. The low-frequency limit may be determined by the decoherence process as the maximum τ will be imposed by the value of T${}_{2}$. These expectations, however, have not been experimentally investigated yet. In this study, we investigate the frequency dependence of AC-field sensitivity using an ensemble of NV centers in diamond. By accounting for the finite widths of π pulses and the duration of optical readout/repolarization, we successfully explain the obtained AC-field sensitivity.

## 2. Results and Discussion

### 2.1. The Effect of Finite-Width π Pulses

In the following two subsections, we provide readers with a concise reminder of how the finite width of π pulses influences the conditions of AC sensing protocols. Furthermore, we highlight the key features of the SR method, which will be exploited to interpret the frequency dependance of the sensitivity. For an oscillating field $B\left(t\right)=B_{ac}\cos\left(2\pi f_{ac}t+\phi_{ac}\right)$ to be commensurate with the π pulses, as illustrated in Figure 1a, the frequency $f_{ac}$ should satisfy the condition below [16,30]
(1)$$f_{ac}=\frac{1}{2\tau (1+\alpha )},$$
where $\alpha =\tau_{\pi }/\tau$ given τ and $\tau_{\pi }$, as illustrated in Figure 1a. For Carl–Purcell type sequences, the accumulated phase Φ can be written as Φ = $\gamma B_{ac}N\tau \left(1+\alpha \right)W$, in which *N* is the total number of the π pulses. According to Ref. [30], the weighting function *W* is given as
(2)$$W\left(\alpha ,\beta ,\phi_{ac}\right)=\frac{\sin(N\beta )}{N\beta }\left[1-\frac{\cos(\beta \frac{\alpha }{1+\alpha })}{\cos(\beta )}\right]\cos\left(N\beta +\phi_{ac}\right),$$
where β is defined as $\pi f_{ac}\tau \left(1+\alpha \right)$. If the frequency $f_{ac}$ satisfies the condition of Equation (Equation 1), β=π/2. Then, α and the phase offset $\phi_{ac}$ crucially affect the accumulated phase Φ. In Figure 1b, the normalized weighting function $\bar{W}\left(\alpha \right)$ (= $\frac{\pi }{2}W\left(\alpha ,\phi_{ac}=0\right)$) is depicted as a function of α. If α<0.2, the reduction in $\bar{W}\left(\alpha \right)$ is less than 4%. As α further increases, $\bar{W}\left(\alpha \right)$ shows a significant decrease, which implies that the DD sequence becomes highly insensitive to AC magnetic fields. In Equation (Equation 2), $\frac{\alpha }{1+\alpha }$ can be substituted by $\frac{f_{ac}}{f_{\mathrm{Rabi}}}$, in which Rabi frequency $f_{\mathrm{Rabi}}$ is defined as $\frac{1}{2\tau_{\pi }}$. As $f_{ac}$ approaches to $f_{\mathrm{Rabi}}$, α becomes infinitely high. This explains that the AC-field sensing protocol is not able to detect the frequencies higher than Rabi frequency ($f_{\mathrm{Rabi}}$).

### 2.2. Sequential Readout Method

The principle of the SR method is illustrated in Figure 2. The unit block of SR consists of a DD sequence for sensing AC fields and laser pulse for optical readout and repolarization. The total duration $T_{\mathrm{SR}}$ splits into the interrogation time, $\tau_{sens}=N\tau \left(1+\alpha \right)$, and the rest δ. The duration of δ includes the laser pulse, $T_{\mathrm{Laser}}$, and a time margin that is practically inevitable. In general, δ may not be commensurate with $B_{ac}\left(t\right)$ as the case in Figure 2. According to Equation (Equation 2), the weighting function depends on the phase $\phi_{ac}$ of the ac field $B_{ac}\left(t\right)$. At the *i*th sampling time t${}_{i}$ (=$iT_{\mathrm{SR}}$), the starting phase can be expressed as $2\pi f_{ac}i\delta$, which is illustrated by the increasing areas in light orange color in Figure 2. Because the SR method entails an undersampling of $B_{ac}\left(t\right)$, the obtained signal S${}_{sens}\left(t\right)$ has a down-converted frequency $f_{sens}$ given as
(3)$$f_{sens}=f_{ac}-\bar{n}f_{\mathrm{SR}},$$
in which the integer $\bar{n}={\mathrm{argmin}}_{n}\left(|f_{ac}-nf_{\mathrm{SR}}|\right)$, and the sampling frequency $f_{\mathrm{SR}}=1/T_{\mathrm{SR}}$ (see the Appendix A for derivation). Equation (Equation 3) implies that the measured frequency $f_{sens}$ is determined by the sampling frequency $f_{\mathrm{SR}}$, irrespective of the DD sequence, but the frequency $f_{ac}$ should be within the sensible bandwidth ($f_{\mathrm{BW}}$ in Figure 1c). Then, the amplitude of the measured signal, $A_{sens}$, can be expressed as
(4)$$A_{sens}=S\cdot C\cdot \sin\left(\Phi_{\left(\phi_{ac}=0\right)}\right).$$

*S* is the intensity of the optical signal acquired during $T_{\mathrm{Laser}}$, and *C* is the contrast induced by spin-manipulating pulses within the DD sequence. The term of sin(Φ) appears when the readout pulse of the DD sequence is $\frac{\pi }{2}\left(Y\right)$, as shown in Figure 2. For a weak AC magnetic field, sin(Φ) can be approximated as Φ.

### 2.3. Bandwidth

We experimentally determined the key features of NV AC-magnetometry using the XY4-(4) sequence. We choose this sequence because it exhibited the highest slope in the variation of the contrast as a function of AC-field amplitude (see the Appendix A). Since the readout pulse is $\frac{\pi }{2}\left(X\right)$, the output signal is proportional to cos(Φ), and the formation of a dip happens when the frequency of external AC field satisfies Equation (Equation 1). In the presence of a 1 MHz AC field with unknown intensities, we scanned τ and found the dips at theoretically expected values (see the Appendix A). For the main peak, τ should be 420 ns because $\tau_{\pi }$ = 80 ns in our experiment. With such τ and $\tau_{\pi }$ (α=0.19), the intensity of the external 1MHz AC field was calibrated. As the amplitude $B_{ac}$ of the AC field increases, the contrast of NV magnetometry follows a periodic oscillation as cos(Φ) curve. The first minimum point corresponds to the phase accumulation of π. From the equation of Φ=π, the amplitude $B_{ac}\left(\pi \right)$ of the AC magnetic signal for the π phase accumulation can be obtained as below
(5)$$B_{ac}\left(\pi \right)=\frac{\pi^{2}f_{ac}}{\gamma N\bar{W}\left(\alpha \right)}.$$

The numerically calculated value of $\bar{W}\left(0.19\right)$ (=0.968) was used for our calibration.

In order to measure the bandwidth of the XY4-(4) sequence, we swept the frequency of AC field across 1 MHz. The obtained data were compared with the theoretical prediction $\cos(\Phi \left(f_{ac}\right))$, where $\Phi \left(f_{ac}\right)=\frac{\gamma NB_{ac}}{\pi f_{ac}^{\left(0\right)}}\bar{W}\left(f_{ac}\right)$. In Equation (Equation 2), $\beta =\frac{\pi }{2}f_{ac}/f_{ac}^{\left(0\right)}$, $f_{ac}^{\left(0\right)}$ = 1 MHz, *N* = 16, and $\phi_{ac}$ = 0. The intensity of the ac field we applied is 1.9 μT. The orange line in Figure 1c is the result of the fitting, where we only used a single parameter to match the height of the dip. The obtained data show good agreement with the theoretical prediction (orange), and the bandwidth $f_{BW}$ is found out to be approximately 50 kHz.

### 2.4. Sensitivity

To measure the sensitivity of AC magnetic fields, we used the SR method, of which main principle is described above (see Appendix B). For a period of 1 s, we acquired the oscillating signal, $S_{sens}\left(t\right)$. In the presence of a reference AC-field having a calibrated rms intensity, the sensitivity can be estimated from the signal to noise ratio (SNR) after Fourier transformation. In the experiment, we used the XY4-(4) sequence to measure the sensitivity. $T_{\mathrm{SR}}$ is set to 50 μs and repeated 20,000 times. The τ value is adjusted for 1 MHz, but the actual frequency of the reference AC-field was slightly detuned by 4 kHz ($f_{ac}$ = 996 kHz) to avoid being commensurate with the sampling rate $f_{\mathrm{SR}}$. The detuning causes insignificant deterioration of the obtained signal because 4 kHz is far less than the bandwidth f${}_{\mathrm{BW}}$ (=50 KHz). The noise floor, which is indicated by the red dashed line below in Figure 3a, can be converted to a detectable magnetic field δB by multiplying the ratio between the rms intensity of the applied magnetic field (0.3211 μT) and the peak intensity at 4 kHz in the Fourier transformed spectrum (0.3211 μT corresponds to 34 mV${}_{\mathrm{PP}}$ shown in Appendix A). Because the total measurement time corresponds to 1 s, the sensitivity η can be obtained as η = δB/$\sqrt{\mathrm{Hz}}$. As shown in Figure 3a, we obtained the sensitivity of 229 pT/$\sqrt{\mathrm{Hz}}$. The main peak appears at 4 kHz, which is consistent with Equation (Equation 3). The 2nd harmonic peak is observed due to the non-linearity in the response to the amplitude of the AC field (refer to Appendix A). Figure 3b shows the time series data of the oscillating signal of the NV fluorescence measure for 1 s, and Figure 3c is its zoomed view for 5 ms.

### 2.5. Frequency Dependence of Sensitivity

We measured the frequency dependence of the sensitivity obtained by the SR method. If the dwell time $T_{\mathrm{SR}}$ is fixed, the sensitivity in Figure 3 will be proportional to the amplitude $A_{sens}$ of the time-domain signal in Equation (Equation 4). In varying the sensing frequency, τ should be adjusted to meet the condition of Equation (Equation 1) for each $f_{ac}$ and thereby β=π/2 in Equation (Equation 2). For producing AC magnetic fields in the experiment, we used a single-turn coil, whose inductance is not negligible. Knowing the AC field intensity at 1 MHz attained from the calibration procedure in Section 2.3, we obtained the actual AC field intensities at other frequencies by comparing the current values (see the current variation in the Appendix A). For theoretical prediction, the parameters *S*, *C*, and Φ in Equation (Equation 4) need to be expressed in terms of $f_{ac}$. $\bar{W}$ is relatively simple because $\frac{\alpha }{1+\alpha }=\frac{f_{ac}}{f_{\mathrm{Rabi}}}$ in Equation (Equation 2) and $\phi_{ac}$ can be set to 0. The contrast *C* in Equation (Equation 4) decays as a function of τ due to decoherence processes like $C\left(\tau /T_{2}\right)=C_{0}\exp\left(-{\left(N\tau /T_{2}\right)}^{p}\right)$, in which *p* and $T_{2}$ are estimated from decoherence curve fitting (see the Appendix A for details). τ varies with $f_{ac}$ following Equation (Equation 1) as τ = $1/2(1/f_{ac}-1/f_{\mathrm{Rabi}})$. Figure 4a shows that the signal intensity *S* increases with $T_{\mathrm{Laser}}$ because the degree of NV spin polarization to the $m_{S}=0$ state becomes higher. This repolarization process can be expressed as $S\left(T_{\mathrm{Laser}}/T_{p}\right)=S_{0}\left(1-\exp\left(-T_{\mathrm{Laser}}/T_{p}\right)\right)$ with the polarization time constant $T_{p}$. The numerical fitting found $T_{p}$ to be 15 μs. Finally, $T_{\mathrm{Laser}}$ is related to $f_{ac}$ through $T_{\mathrm{SR}}$. The relationship is described by $T_{\mathrm{Laser}}$ = $T_{\mathrm{SR}}$−Nτ(1+α) = $T_{\mathrm{SR}}$−$N/2f_{ac}$. Including them altogether, we can describe how the sensitivity varies as a function of $f_{ac}$,
(6)$$\eta \left(f_{ac}\right)\propto \frac{1}{S\left(T_{\mathrm{Laser}}/T_{p}\right)C\left(\tau /T_{2}\right)\Phi \left(f_{ac}\right)}\sqrt{\frac{T_{\mathrm{SR}}}{N\tau (1+\alpha )}}.$$

The first terms originate from the inverse of the amplitude in Equation (Equation 4), and the last term represents the effect of the overhead time [22,24]. The theoretical prediction is shown as an orange curve in Figure 5a. The experimental data in Figure 5b show good agreement with it. The high-frequency limit is clearly determined by the Rabi frequency, being approximately 6.3 MHz in the present study, which is indicated by the red dashed line. The low-frequency limit is close to 200 kHz (yellow dashed line). We found that the effect of decoherence causing the decay of the contrast $C(\tau /T_{2})$ alone cannot explain it. The blue curve in Figure 5a is obtained by omitting $S(T_{\mathrm{Laser}}/T_{p})$ from Equation (Equation 6), and its low-frequency limit is below 100 kHz, indicated by blue dashed line. This discrepancy originates from the reduction in optical signal intensity *S* when the duration of T${}_{\mathrm{Laser}}$ decreases (Figure 4a). Given T${}_{\mathrm{SR}}$, lowering $f_{ac}$ accompanies decreasing T${}_{\mathrm{Laser}}$, as seen by the fitting equation $T_{\mathrm{Laser}}$ = $T_{\mathrm{SR}}$−$2N/f_{ac}$ in Figure 4b. The reduced signal intensity results in the deterioration of the sensitivity. In addition, similar to previous works [22,24], the theoretical prediction described by Equation (Equation 6) has the optimal frequency, which is found to be within the range from 800 kHz to 1 MHz. We achieved a sensitivity of 229 pT/$\sqrt{\mathrm{Hz}}$ at 1 MHz (Figure 3a).

In Equation (Equation 6), the term $S(T_{\mathrm{Laser}}/T_{p})$ sets it apart from earlier theoretical predictions [22,24]. Governed by the ratio $T_{\mathrm{Laser}}/T_{p}$, mitigating the impact of $S(T_{\mathrm{Laser}}/T_{p})$ and thus reaching the lower frequency limit determined by $T_{2}$ can be accomplished through two approaches. The first method is associated with the optical power density irradiated to NV centers. The power density for optical readout/repolarization should be high enough to sufficiently polarize NV spins within $T_{\mathrm{Laser}}$. With a laser spot diameter of 0.2 mm and laser power of 1.4 W in this work, the power density utilized is approximately 45 W/mm${}^{2}$. However, this value still falls significantly short of the saturation density of NV spins (>5 kW/mm${}^{2}$) as estimated from single NV experiments [14]. Given the considerably high saturation power density for NV spins, practical challenges might arise in achieving complete optical repolarization during the $T_{\mathrm{Laser}}$ time with an ensemble of NV centers. The second method involves increasing $T_{\mathrm{Laser}}$. This leads to the increase in the dwell time $T_{\mathrm{SR}}$ of the SR method. Essentially, the SR method emulates the acquisition of a highly under-sampled signal as shown in Figure 2. If a sinusoidal signal in the time domain is subjected to a certain amount of Gaussian noise, the noise floor of the under-sampled signal decreases at a higher sampling rate. This effect stems from the fact that higher sampling rates distribute the noise across a wider frequency. Hence, during the measurements for Figure 5b, we maintained a fixed value for $T_{\mathrm{SR}}$ to prevent variations in the noise floor due to changes in bandwidth. When employing sequential readout, the selection of $T_{\mathrm{SR}}$ is imperative in light of the desired bandwidth. What Equation (Equation 6) implies is that the consideration of $T_{\mathrm{Laser}}$ and $T_{p}$ plays an essential role in anticipating the lower frequency limit.

## 3. Conclusions

In this work, we present a combination of experimental and theoretical investigations into the frequency-dependent behavior of AC magnetic field sensitivity through the use of sequential readout. Our findings unveil that the upper limit is determined by the finite-width of π pulses, while the lower limit can be governed by the duration of optical repolarization for NV ensembles when the optical power density is often below the saturation limit. With the dwell time of the SR method held constant, our theoretical model, which encompasses these two factors, demonstrates a reasonable concordance with the experiment results. Furthermore, through the employment of the XY4-(4) DD sequence, we obtained a maximum sensitivity of 229 pT/$\sqrt{\mathrm{Hz}}$ at 1 MHz. Considering the sensitivity reported in the previous study [1], ranging from 30 to 70 pT/$\sqrt{\mathrm{Hz}}$, we hold the conviction that the achieved sensitivity empowers us to conduct NV-NMR in micron-scale, where (thermal) Boltzmann polarization remains predominant [31]. This work not only offers valuable insight but also furnishes practical guidelines for the applications of AC magnetometry using NV spins in the range of kilohertz to megahertz. These applications may include other areas, such as magnetic induction tomography, in which NV spins are expected to facilitate high spatial resolution imaging.

## Figures and Tables

**Figure 1 sensors-23-07566-f001:**
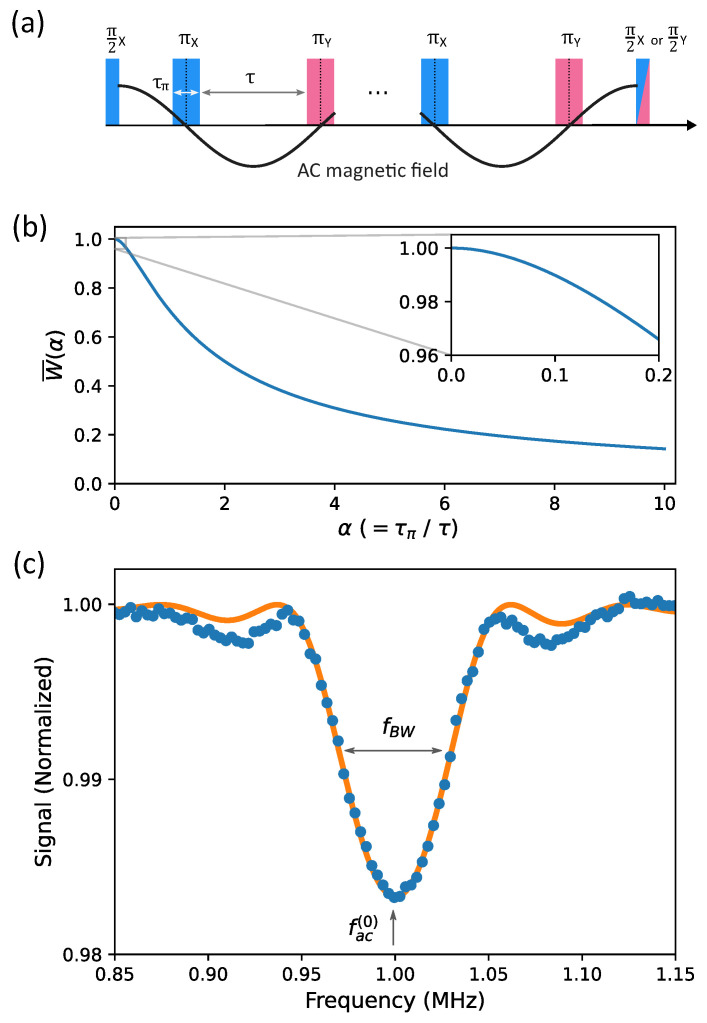
(**a**) The sequence of XY4-(k) in the presence of an AC magnetic field commensurate with the π pulses. (**b**) The normalized weighing function $\bar{W}$ (see the main text) is plotted as a function of α (=$\tau_{\pi }/\tau ).$ (**c**) The bandwidth curve of the AC sensing employing XY4-(4) is shown at the center frequency ($f_{ac}^{\left(0\right)}$) of 1 MHz. The theoretical curve fitted with only a single parameter to match the dip height is shown in orange. The bandwidth $f_{BW}$ is approximately 50 kHz.

**Figure 2 sensors-23-07566-f002:**
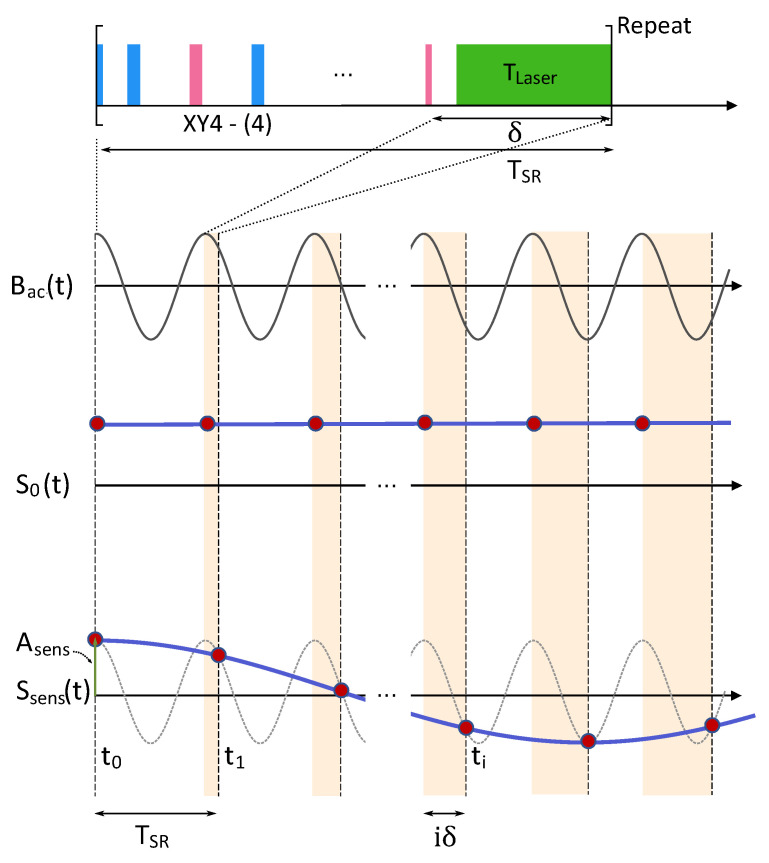
The principle of sequential readout is illustrated. The top exhibits the unit block and its duration is $T_{\mathrm{SR}}$. The unit block includes DD (XY4-(4)) sequence and optical readout/repolarization. If $T_{\mathrm{SR}}$ is commensurate with the period of $B_{ac}\left(t\right)$, a constant signal $S_{0}\left(t\right)$ will be observed. If not, an under-sampled signal $S_{sens}\left(t\right)$ is acquired and its frequency is given by Equation (Equation 3).

**Figure 3 sensors-23-07566-f003:**
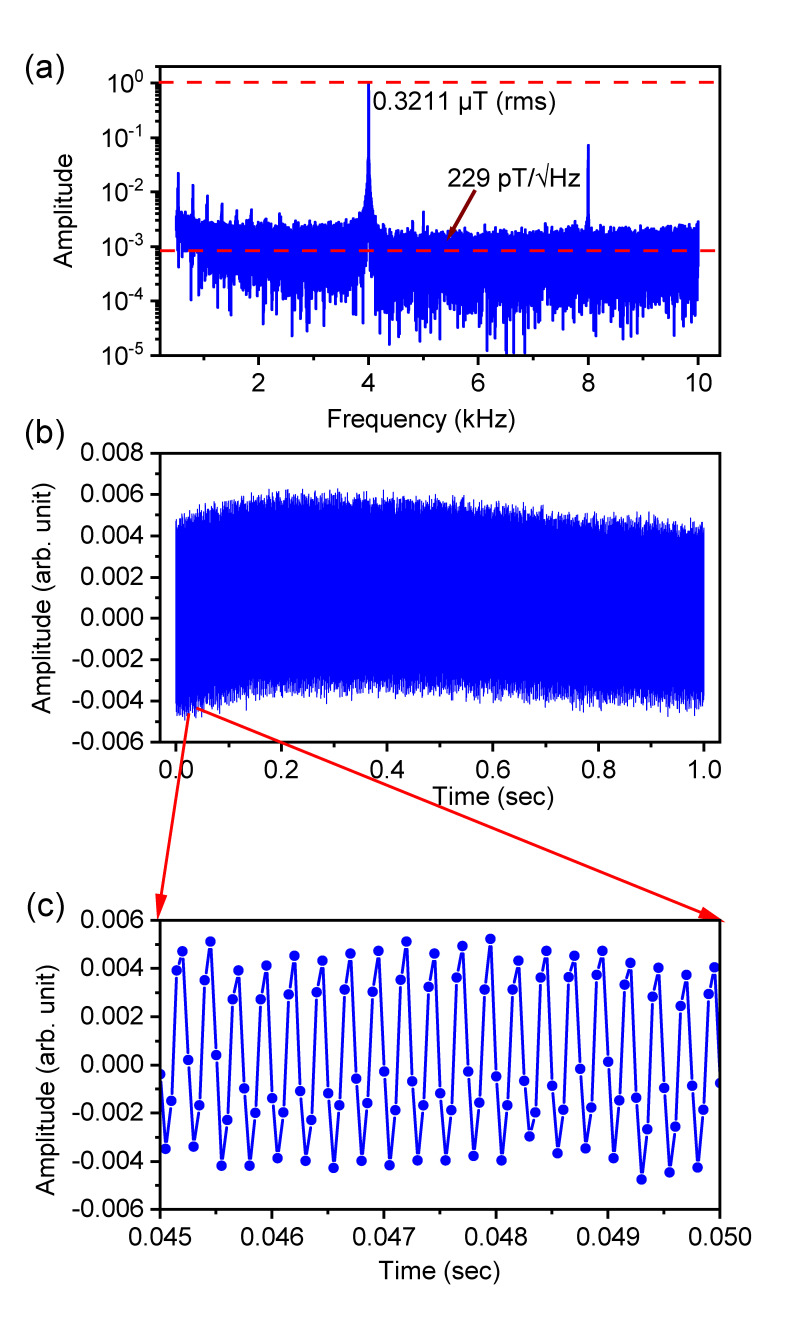
(**a**) Fourier transform of the time-domain signal $S_{sens}\left(t\right)$ obtained by the SR method. The rms noise floor is calibrated from the intensity of the applied AC field, 0.3211 μT (rms), and the sensitivity is 229 pT/$\sqrt{\mathrm{Hz}}$. (**b**) The time-domain data of the oscillating signal of NV fluorescence measured for 1 s, and (**c**) its zoomed view for 5 ms. The amplitude of the obtained signal corresponds to a contrast of approximately 1%. In (**b**,**c**), dc baseline offset is eliminated.

**Figure 4 sensors-23-07566-f004:**
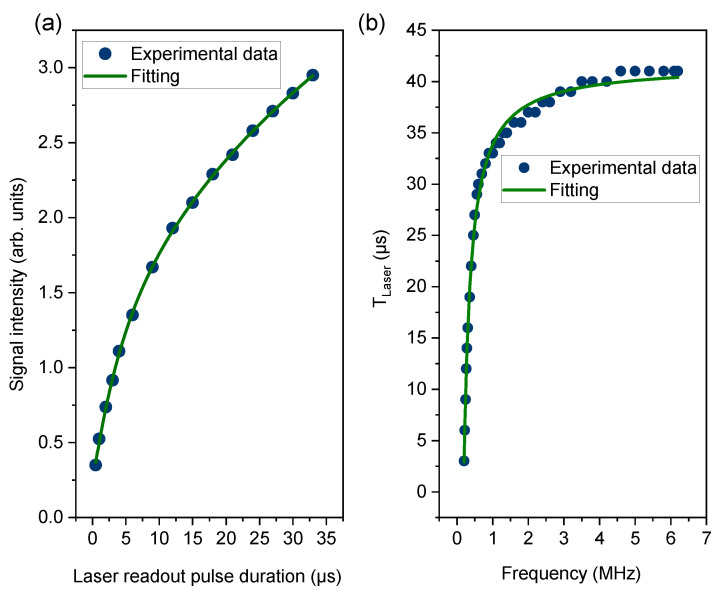
(**a**) NV fluorescence intensity is measured as a function of optical readout duration. The repolarization time $T_{p}$ is estimated to be 15 μs from exponential fitting. (**b**) The duration T${}_{\mathrm{Laser}}$ of optical readout and repolarization varies according to the sensing frequency because the dwell time T${}_{\mathrm{SR}}$ is fixed. The values of T${}_{\mathrm{Laser}}$ used in a series of measurement nearly follow the relation, $T_{\mathrm{Laser}}$ = $T_{\mathrm{SR}}$−$N/2f_{ac}$.

**Figure 5 sensors-23-07566-f005:**
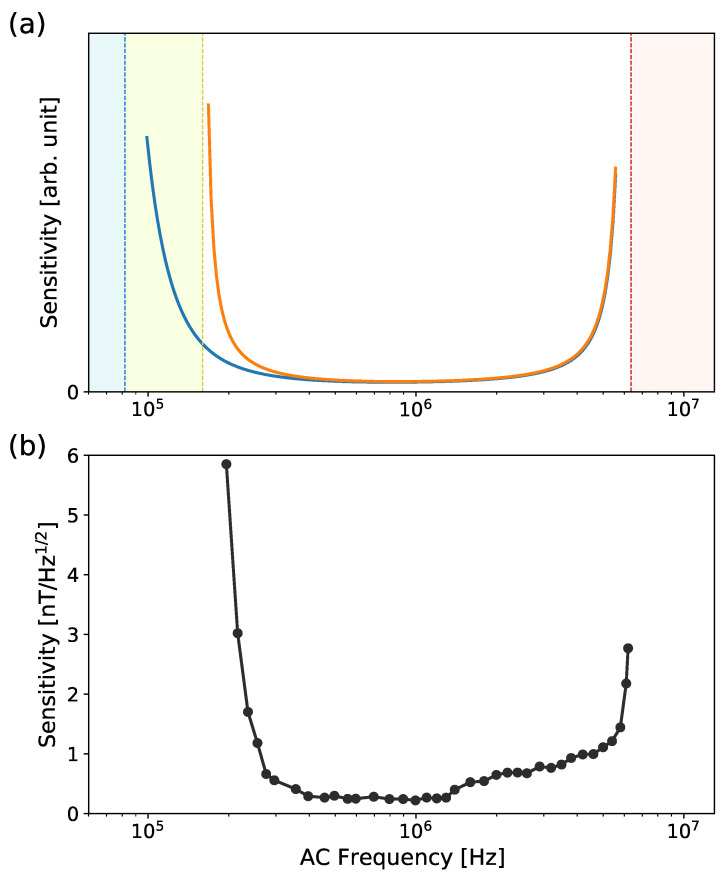
The frequency dependence of AC sensitivity based on the SR method is investigated. (**a**) Theoretical prediction according to Equation (Equation 6) is plotted in orange color. The red dashed line on the left side represents Rabi frequency limit (6.3 MHz). The lower limit is shown in orange dashed line (160 kHz). The blue dashed line (82 kHz) indicates T${}_{2}$ limit, which is obtained by omitting $S(T_{\mathrm{Laser}}/T_{p})$ from Equation (Equation 6). (**b**) Experimentally obtained AC sensitivity as a function of frequency is in accordance with theoretical prediction.

## Data Availability

Not applicable.

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
