# Peer review of "Frequency Limits of Sequential Readout for Sensing AC Magnetic Fields Using Nitrogen-Vacancy Centers in Diamond"

_sensors, 2023, doi:10.3390/s23177566_

Round 1
Reviewer 1 Report
The article explores the bandwidth/frequency range of AC magnetic field sensing using NV centers in diamond, based on the Sequential Readout method utilizing Dynamic Decoupling pulse sequences. The key novelty highlighted by the authors is that the lower frequency limit is governed by the duration of the optical repolarization pulse, rather than the coherence time T2. Sounds interesting, however I would like to ask the following questions and suggest some changes:
Questions
Q1. The authors claim that the lower frequency limit is governed by the optical readout laser pulse. However, this is true only when the dwell time (TSR) of the sequential readout is fixed in order to maintain a constant bandwidth. In order to keep TSR fixed, TLaser needs to be reduced when lower frequencies have to be measured, thereby degrading the sensitivity.
I am curious to know what happens if the bandwidth is kept flexible (can be lowered), and TLaser is kept fixed. In this case, the repolarization pulse width will not influence the sensitivity, so will the lower frequency limit be decided by the coherence time (T2)?
The authors should explicitly state that their hypothesis is for the specific case when the acquisition bandwidth is fixed. They should provide reasons/advantages of keeping the bandwidth fixed. They briefly mention that the noise floor will be affected by the bandwidth, however you will gain sensitivity as well since TLaser won’t have to be reduced.
Q2. The curve shown in Fig. 5 (a) between signal intensity and laser readout pulse duration is extremely important, since this explains why the sensitivity degrades when TLaser reduces. They claim that the signal intensity increases with TLaser, because the degree of NV spin polarization to mS = 0 state becomes higher. However, does such a curve/mathematical relationship hold for single NV centers as well? Or it is the case for only NV ensembles?
Please take a look at Fig 4 (c) in (Annual Review of Physical Chemistry 2014 65:1, 83-105). The intensity obtained for single NV center doesn’t monotonically increase with pulse width, if I am interpreting it correctly.
Please clarify if this relationship holds for NV ensembles only, or its true for single NV centers as well. This is quite important, since NMR measurements are mostly performed with single NV centers.
Q3. The authors claimed that the NV saturation limit hasn’t been reached. Does this imply that frequencies lower than 160KHz (lower limit calculated) can be measured if the laser power is further increased?
Q4. A maximum sensitivity of 229 pT/√ Hz by employing the XY4-(4) 10 DD sequence. The authors did not compare this obtained sensitivity with previous obtained values. Is it comparable, significantly better or worse? Is this sensitivity good enough for micron-scale NMR applications, as they claimed?
Q5. The authors didn’t explain the detailed procedure used to calculate the magnetic field sensitivity (229 pT/√ Hz). Please add it to the Supporting Information, or cite an appropriate reference.
Q6. For the sensitivity calculation, TSR is 50 µs, however T2 is calculated to be 16 µs in the SI. How can the dwell time be higher than the T2 time?
Q7. Overall, the authors have described the technical aspects of this work in a very detailed manner (experiments, mathematical relations etc.). However, I would suggest adding a few sentences the potential practical applications of this work and how it helps the field. For eg. What are the use cases of KHz range AC magnetic field sensing.
Q8. What is the contrast (%) between maximum and minimum NV fluorescence in Fig. 3©?
1. Line 136: Figure 5(a).
2. Cite reference in line 36 for Sequential Readout, QDyne, and projective readout. Ref. 25 is just a short summary. I would suggest citing Refs. 26-27.
3. Line 40: solution resolution
4. Throughout the main text, please ensure every equation is either obtained from a cited reference, or explained in detail in the Supporting Information. For eg. Equations (3) and (4) did not have citations.
5. The increasing areas in light green color in Fig. 2 can be a little misleading, since it seems to suggest the readout pulse width is increasing. I would suggest changing the color maybe to avoid the confusion.
6. Fig. 3 ©. The authors should explain how they calculate the arbitrary units for y axis. For a general reader, it can be confusing to see the NV fluorescence be negative.
Reviewer 2 Report
The paper by Santosh Ghimire et al. deals with a current hot topic - quantum sensing with NV centres in diamond. In particular, the authors explore the frequency limit of a quantum sensing protocol that allows high-frequency resolution and was first demonstrated a few years ago. The finding itself is neither surprising nor entirely new, but the results are helpful to researchers in the field, in particular to those, who enter the field. However, before the paper can be published, the following questions need to be answered:
Major:
1) Could the authors check the alpha factor? In Figure 1b, the weighting function goes down with larger alphas (alpha = tau pi/tau). This means that alpha becomes large when the distance between pi pulses becomes large. Does this make sense?
2) Calibration: How much Vpp corresponds to the 0.32 uT? Are the authors still in the linear regime of the NV response? Also, the authors use different units for the calibration signal (Vpp, RMS...). Please unify.
3) Did the authors calibrate the signal for the different frequency ranges they used? The coil response can be very different in the range between 200 kHz and 5 MHz.
4) How did the authors determine their T2? The fitting function in the SI lacks the number of pi pulses. Please clarify.
Minor:
5) Could the authors clarify how they detune the frequency (nfSR or tau)?
6) How exactly did the authors calculate their SNR?
7) Equation 6: Can the authors use T2 in this equation – the reader will be looking for this number. Currently, it is hidden in the contrast as a function of tau.
8) There are many abbreviations in Fig. S1 that need to be defined.
9) Please proofread the SI. There are many typos.
